# Spirometry Examination of Adolescents with Thoracic Idiopathic Scoliosis: Is Correction for Height Loss Useful?

**DOI:** 10.3390/jcm10214877

**Published:** 2021-10-22

**Authors:** Katarzyna Politarczyk, Mateusz Kozinoga, Łukasz Stępniak, Paweł Panieński, Tomasz Kotwicki

**Affiliations:** 1 Department of Spine Disorders and Pediatric Orthopaedics, University of Medical Sciences, 61-545 Poznan, Poland; mkozinoga@hotmail.com (M.K.); lukaszstepniak22@gmail.com (Ł.S.); kotwicki@ump.edu.pl (T.K.); 2 Department of Emergency Medicine, University of Medical Sciences, 60-355 Poznan, Poland; ppanienski@ump.edu.pl

**Keywords:** idiopathic scoliosis, body height, pulmonary function test, Cobb angle

## Abstract

Loss of body height is observed in patients with idiopathic scoliosis (IS) due to spine curvatures. The study compared pulmonary parameters obtained from spirometry examination considering the measured versus the corrected body height. One hundred and twenty adolescents with Lenke type 1 or 3 IS who underwent preoperative spirometry examination and radiographic evaluation were enrolled. The mean thoracic Cobb angle was 68° ± 12.6, range 48–102°. The difference between the measured and the corrected body height increased with the greater Cobb angle. Using the corrected body height instead of the measured body height significantly changed the predicted values of pulmonary parameters and influenced the interpretation of the pulmonary testing results.

## 1. Introduction

Spine and trunk alignment can be altered due to idiopathic scoliosis (IS), and this can impair pulmonary function in the case of spinal curvatures developing in the thoracic region [1]. Previous publications regarding possible factors contributing to pulmonary function impairment revealed that radiological parameters such as thoracic Cobb angle, thoracic kyphosis angle, the number of vertebrae involved and the limitation of rib cage mobility might correlate with the pulmonary parameters [2,3,4,5,6]. Due to spine deformity, a loss of body height is observed in patients with idiopathic scoliosis, which can be considered another factor which may impact pulmonary testing results. In previous studies, several mathematical formulas for calculating scoliosis-induced body height loss were proposed, but none are considered a gold standard [7,8,9,10,11]. No study comparing the impact of the application of the measured body height versus the corrected body height (defined as the sum of the measured body height and the body height loss caused by the spinal deformity) on the interpretation of the spirometry testing results was identified.

The aim of the study was to compare the pulmonary parameters revealed during spirometry examination in adolescents with thoracic idiopathic scoliosis in relation to measured versus corrected body height. The hypothesis is that using the patients’ measured body height introduces a bias in the interpretation of the spirometry examination in patients with thoracic idiopathic scoliosis.

## 2. Materials and Methods

### 2.1. Study Population

Retrospective analysis of preoperative radiographs and preoperative pulmonary testing was performed in 120 adolescents (88 females and 32 males) aged 15.0 ± 1.8 years (range 12–18), who were admitted for surgical IS treatment and met the inclusion criteria: diagnosis of IS, Lenke 1 or 3 type curve, surgical treatment, spirometry evaluation. The exclusion criteria comprised non-idiopathic scoliosis and previous surgical treatment. The Lenke 1 group consisted of 73 patients (53 girls and 20 boys), the Lenke 3 group of 47 patients (35 girls and 12 boys).

The patients’ charts were analyzed according to the Lenke curve type (Lenke 1 vs. Lenke 3) and according to the thoracic Cobb angle: subgroup I. of 84 patients with thoracic Cobb angle less than 75° vs. subgroup II. of 36 patients with thoracic Cobb angle of 75° or more.

### 2.2. Radiological Examination

In accordance with the Cobb method [12], the thoracic and lumbar curve measurements were taken from the standing anteroposterior radiograph of the whole spine.

### 2.3. Corrected Body Height Calculation

The loss of body height was calculated on the basis of Stokes’ formula [10]. The corrected body height was calculated as the sum of the measured body height and the loss of body height. In the case of single scoliosis (Lenke 1), the formula for calculating the loss of body height was 1.55 − 0.0471Cobb + 0.009Cobb^2^, while in the case of double curves (Lenke 3) the formula was 1.0 + 0.066Cobb + 0.0084Cobb^2^ [10].

### 2.4. Pulmonary Testing

Pulmonary testing (PT) was performed in a sitting position using a LungTest LT 250 spirometer (MES, Kraków, Poland). Forced Vital Capacity (FVC) and Forced Expiratory Volume in one second (FEV1) were measured 3 times. The single best effort was taken for analysis [5,13,14,15,16]. The predicted values of the pulmonary parameters, the lower limit of normal (LLN), the upper limit of normal (ULN), z-scores and percentages of the predicted values of the pulmonary parameters were calculated according to the Global Lung Function Initiative (GLI 2012) reference values, independently for the measured and the corrected body height [17].

### 2.5. Statistical Analysis

The mean value, standard deviation and range of the parameters were calculated using Microsoft Excel Software (Microsoft, Redmond, WA, USA). The Kolmogorov–Smirnov test was used to analyze normal data distribution. Since the parameters revealed normal distribution, Student’s *t*-test was used to determine the significance of the differences for predicted values LLN, ULN, z-scores and percentages of the predicted values of the pulmonary parameters calculated according to the measured and the corrected body height. The correlation between the thoracic Cobb angle magnitude versus the loss of body height was calculated using Pearson’s correlation coefficient.

The statistical significance level was set at *p* = 0.05. The analysis was performed with Statistica Software (TIBCO Software Inc., Palo Alto, CA, USA).

## 3. Results

### 3.1. Cobb Angle Analysis

The mean thoracic Cobb angle was 68° ± 12.6 (48–102°) for all patients. For Lenke 1 subgroup I, the thoracic Cobb angle was 60.2° ± 6.6 (48–74°); for Lenke 1 subgroup II, the thoracic Cobb angle was 82.6° ± 6.7 (75–102°). In Lenke 3 subgroup I, the thoracic Cobb angle was 63.8° ± 6.0 (50–74°); in Lenke 3 subgroup II, the thoracic Cobb angle was 87.3° ± 6.2 (76–95°).

### 3.2. Measured versus Corrected Body Height

The values of the measured body height, the calculated corrected body height and the calculated loss of body height are presented in Table 1.

The body height loss was significantly higher in subgroup II patients than in subgroup I (5.7 cm vs. 3.1 cm, *p* < 0.01). For the Lenke 1 and Lenke 3 type considered separately, body height loss was also higher in subgroup II than in subgroup I (5.9 cm vs. 3.2 cm, *p* < 0.01; 5.3 cm vs. 3.1 cm, *p* < 0.01, respectively).

With the increasing Cobb angle, increased body height loss was observed, as presented in Figure 1.

### 3.3. Predicted Pulmonary Parameters Calculated for the Measured versus the Corrected Body Height

The predicted absolute value of FVC and FEV1, as well as the values of the LLN and ULN for each of the two pulmonary parameters (FVC and FEV1), were calculated according to the GLI 2012 reference values [17], using the measured body height and using the corrected body height, separately (Table 2), as well as within the Lenke types and the subgroups (Appendix A). All the corresponding values—the predicted values of the FVC and FEV1, LLN and ULN calculated for the FVC and FEV1—proved significantly different.

### 3.4. Pulmonary Parameters Values Registered at Spirometry Examination

The absolute FVC and FEV1 values, followed by z-score and the percentage of the predicted normal value for FVC and FEV1, are presented in Table 3. Z-score and the percentage of the predicted normal value for FVC and FEV1 were calculated for the measured versus the corrected body height.

### 3.5. Comparison of the Pulmonary Parameters in Subgroup I versus Subgroup II

The predicted absolute value of FVC and FEV1, as well as the values of LLN and ULN for each of the pulmonary parameters (FVC, FEV1), were calculated using the measured body height and using the corrected body height in subgroup I and subgroup II in Lenke 1 and Lenke 3 types, separately (Table A1), in Lenke 1 type (Table A2) and in Lenke 3 type (Table A3). All the corresponding values- the predicted values of the FVC and FEV1, LLN, and ULN calculated for the FVC and FEV1- proved significantly different.

The absolute FVC and FEV1 values, followed by z-score and the percentage of the predicated value of FVC and FEV1 for both subgroups calculated for the measured versus corrected body height, are presented in Table A4.

The FVC absolute values were not significantly different in both subgroups (*p* = 0.14); however, the absolute value of FEV1 was significantly lower in patients with a greater Cobb angle (*p* = 0.04).

The %FVC calculated for the corrected body height and %FEV1 values calculated for the measured and the corrected body height were significantly lower in subgroup II than in subgroup I (*p* = 0.02; *p* = 0.04, *p* = 0.01, respectively). Additionally, the FVC and FEV1 z-score values calculated for the corrected body height were significantly lower in subgroup II than subgroup I (*p* = 0.01; *p* = 0.01, respectively).

The absolute FVC and FEV1 values, as well as the z-score values and the percentages of the predicted values of both parameters calculated in subgroup I and subgroup II using the measured versus corrected body height in Lenke 1 type patients are presented in Table A5.

The absolute values of FVC and FEV1 were lower in subgroup II than in subgroup I patients, but the difference was not significant (*p* = 0.58; *p* = 0.82, respectively).

Significantly lower values of %FVC and %FEV1 were observed in subgroup II patients than in subgroup I when calculated for both the measured and the corrected body height values (*p* = 0.04; *p* = 0.009; *p* = 0.02; *p* = 0.006, respectively). Additionally, FVC and FEV1 z-score values proved significantly lower in patients with a greater Cobb angle when calculated using the measured and the corrected body height (*p* = 0.04; *p* = 0.008; *p* = 0.03; *p* = 0.008, respectively).

The absolute FVC and FEV1 values, as well as the z-score values and the percentages of the predicted values of both parameters calculated in subgroup I and subgroup II using the measured versus corrected body height in Lenke 3 type patients are presented in Table A6.

In Lenke 3 type, the differences in absolute FVC, FEV1 values were not significant in subgroup II versus subgroup I (*p* = 0.20; *p* = 0.13, respectively).

Additionally, in Lenke 3 type, the values of the %FVC and %FEV1 parameters were lower in subgroup II than in subgroup I when calculated for the measured and the corrected body height; however, the differences were not significant (*p* = 0.89; *p* = 0.56; *p* = 0.81; *p* = 0.52, respectively). Additionally, the FVC and FEV1 z-score values proved not significantly different when comparing subgroup I versus subgroup II (*p* = 0.84; *p* = 0.51; *p* = 0.85; *p* = 0.85, respectively).

An example of a different interpretation of the result of spirometry testing is shown in [Fig jcm-10-04877-g0A1].

## 4. Discussion

Spirometry is an examination that allows us to assess pulmonary function in patients with IS. Regular pulmonary testing allows to recognize pulmonary impairment, even though it may not be clinically evident until severe or irreversible changes have occur [18]. Even though spirometry cannot replace body plethysmography for diagnosing the restrictive patterns that occur in patients with idiopathic scoliosis, it may indicate their presence and suggest a need for further examination [19,20]. On the other hand, spirometry is recommended for diagnosing the obturation patterns which are less common in IS [19,20,21,22]. Loss of body height is observed in all patients with moderate or severe idiopathic scoliosis—the greater the Cobb angle, the higher the body height loss. Several methods have been proposed to replace the misleadingly low measured body height during pulmonary function testing.

Hepper et al. [23] observed that body height is correlated with arm span. The authors evaluated pulmonary parameters in patients with kyphoscoliosis using body height calculated from arm span. Using arm span instead of measured body height was recommended for calculating pulmonary parameters in patients with spinal deformities [24]. However, the arm span: body height ratio depends on age and gender [24,25,26]. Furthermore, arm span may be lower than predicted due to the trunk asymmetry caused by spinal deformity [27].

Several factors were previously identified as determinants contributing to body height loss due to idiopathic scoliosis: Cobb angle magnitude, curvature length and the number of vertebrae involved in the curve [7,8,9,10]. In previous studies, authors presented regression equations based on the Cobb angle values that may be used to calculate body height loss [7,8,9,10,28].

Tyrakowski et al. [11] compared four methods—Bjure, Stokes, Kono and Ylikoski—and concluded that none of them could be recommended as most valid. On the other hand, Gardner et al. [29], comparing five methods—Bjure, Stokes, Kono, Ylikoski and Hwang—concluded that the Kono and Stokes methods were the most valid in determining height loss in patients with idiopathic scoliosis.

Our preliminary study in 39 IS patients (29 girls and 10 boys) aged 12–17 with the mean thoracic Cobb angle 69.8° ± 12.4° (50–104°) showed that corrected body height was significantly higher than measured body height (*p* = 0.01) [30]. The impact of body height loss on the interpretation of the spirometry testing is based on the fact that the predicted reference values are calculated for every individual patient considering age, gender, body height and ethnicity. From these data, the predicted values are generated by means of the GLI 2012 regression equations using spirometry equipment software or the GLI 2012 software. Changing one component value—body height—the predicted values of the pulmonary parameters change accordingly. For example, in a 13-year-old girl, the application of the corrected body height (172.2 cm) instead of the measured one (168.0 cm) into the regression equation results in a 5.46% difference in the FVC predicted value (3.89 L vs. 3.67 L) and 5.26% difference in the FEV1 predicted value (3.40 L vs. 3.23 L; Appendix B).

Weinstein et al. [2] concluded that pulmonary parameters—FVC and FEV1—significantly correlate with the thoracic Cobb angle magnitude. The pulmonary parameters decreased when the thoracic Cobb angle value got close to 100°–120°. However, later observations suggested that pulmonary impairment was observed in patients with smaller curves [3,5,31]. In our study, the absolute value of FEV1 (*p* = 0.04) but not FVC (*p* = 0.16) was significantly lower in patients with a greater Cobb angle (Cobb > 75°). Meanwhile, when the spirometry absolute values are transformed into percentages of the predicted values, the impact of the body height correction appears evident. In adolescents with a Cobb angle > 75°, the %FVC_c_, %FEV1_m_ and %FEV1_c_ values were lower than in those with a Cobb angle < 75° (*p* = 0.02; *p* = 0.04, *p* = 0.01, respectively).

In clinical practice, the absolute values of spirometry parameters are less useful for evaluating the patient’s pulmonary status, so clinicians rely more on the percentages of the values predicted for a given gender, age and height. The calculation of the predicted values is automatically offered by spirometry equipment software. The modification of one component—body height—does impact the calculation of the predicted values. In consequence, one absolute value measured in liters may be interpreted differently due to different reference values. This study confirmed the discrepancy of the predicted FVC and FEV1 values depending on the height parameter introduced for calculation (measured vs. corrected). As body height loss increases with the Cobb angle, the discrepancy becomes significant in severe curves, over 75 Cobb degrees.

In previous studies of IS patients, the pulmonary parameters used to be interpreted using the threshold of 80% as the lower limit of normal predicted value, in accordance with the recommendation made by Bates and Christie [32]. The American Thoracic Society and European Respiratory Society recommend the 5th percentile as an LLN (z-score −1.64) [21,33,34]. The z-score value indicates by how many standard deviations a particular measurement is situated from the predicted value. Contrary to the percentage of the predicted value, the z-score parameter is not biased due to age, gender or ethnic group, and seems more useful for defining the LLN [17]. In Lenke 1 type subgroup I, the z-score values of FVC and FEV1 calculated for the measured body height indicated that the results were within the normal limits. However, when the z-score values of FVC and FEV1 were calculated for the corrected body height, the results were below the normal limits (−1.54 vs. −1.84; −1.51 vs. −1.78, respectively). Replacing the measured body height with the corrected body height can also change the classification of the severity of the pulmonary impairment. In subgroup II, the %FVC and %FEV1 calculated from the measured body height indicated a mild pulmonary impairment, but when the corrected body height was used, the interpretation changed to a moderate impairment [20].

A limitation of the study was that the study group consisted solely of patients with Lenke 1 or 3 type scoliosis. Another limitation is that the group contained both girls and boys.

## 5. Conclusions

Increasing height loss is observed with an increasing Cobb angle in adolescents with thoracic idiopathic scoliosis. Predicted spirometry reference values change significantly in proportion to a change in body height. At spirometry examination, the interpretation of the results of the pulmonary functional testing is affected by body height loss. Use of the corrected body height instead of the measured one can be seriously considered in severe thoracic curvatures over a 75 degree Cobb angle.

## Figures and Tables

**Figure 1 jcm-10-04877-f001:**
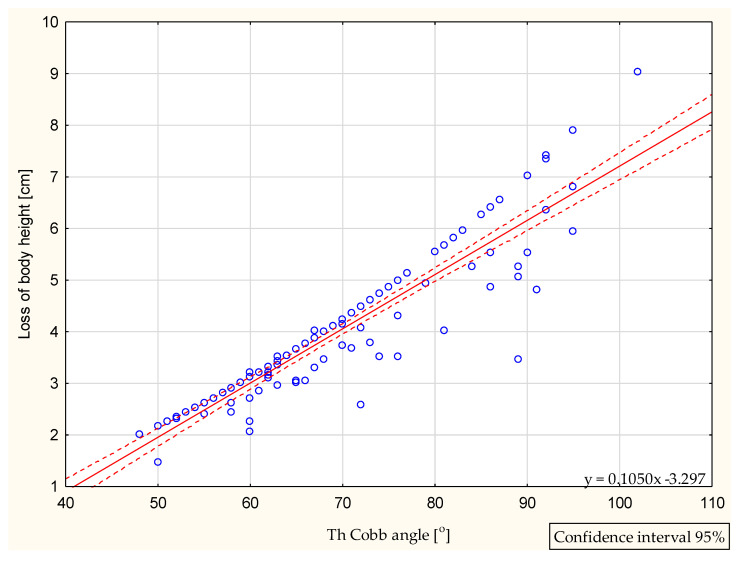
Thoracic Cobb angle magnitude versus loss of body height, Pearson’s correlation coefficient r = 0.93, *p* < 0.01, *N* = 120.

**Table 1 jcm-10-04877-t001:** Comparison of the measured body height versus the calculated corrected body height.

	Thoracic Cobb Angle Range [°]	Measured Body Height [cm]	Calculated Height Loss [cm] (Stokes)	Calculated Corrected Body Height [cm]	Measured versus Corrected Body Height
Lenke 1 and 3 types					
Subgroup I and II *N* = 120	48–102	164.9 ± 7.9 (145.0–185.0)	3.9 ± 1.4 (1.5–9.0)	168.8 ± 8.0 (151.3–188.5)	*p* < 0.01 *
Subgroup I *N* = 84	48–74	164.9 ± 7.9 (148.0–184.0)	3.1 ± 0.7 (1.5–4.7)	168.0 ± 7.8 (151.2–187.5)	*p* = 0.01 *
Subgroup II *N* = 36	75–102	165.0 ± 8.1 (145.0–185.0)	5.7 ± 1.2 (3.5–9.0)	170.7 ± 8.2 (151.3–188.5)	*p* = 0.004 *
Lenke 1 type					
Subgroup I and II *N* = 73	48–102	165.1 ± 7.7 (145–184)	3.9 ± 1.4 (2.0–9.0)	169.0 ± 7.7 (151.2–188.0)	*p* < 0.01 *
Subgroup I *N* = 54	48–74	165.5 7 ± 9 (151.0–184.0)	3.2 ± 0.7 (2.0–4.7)	168.6 ± 7.8 (153.3–187.2)	*p* < 0.01 *
Subgroup II *N* = 19	75–102	163.9 ± 7.2 (145.0–182.0)	5.9 ± 1.0 (4.8–9.0)	169.9 ± 7.2 (151.2–188.0)	*p* < 0.01 *
Lenke 3 type					
Subgroup I and II *N* = 47	50–95	164.7 ± 8.3 (148.0–185.0)	3.9 ± 1.4 (1.5–7.9)	168.6 ± 8.6 (151.5–188.5)	*p* < 0.01 *
Subgroup I *N* = 30	50–74	163.8 ± 7.8 (148.0–184.0)	3.1 ± 0.6 (1.46–4.1)	166.9 ± 7.8 (151.5–187.5)	*p* < 0.01 *
Subgroup II *N* = 17	76–95	166.3 ± 9.0 (149.0–185)	5.3 ± 1.2 (3.5–7.9)	171.6 ± 9.3 (152.5–189.6)	*p* < 0.01 *

All values are presented as a mean ± standard deviation, minimum and maximum in brackets. * difference statistically significant.

**Table 2 jcm-10-04877-t002:** Predicted values of the pulmonary parameters calculated (GLI 2012, [17]) for the measured versus the corrected body height, *N* = 120.

Parameter	FVC_m_ Measured Body Height	FVC_c_ Corrected Body Height	FVC_m_ vs. FVC_c_ *p*-Value
Predicted value [L]	3.83 ± 0.6	4.04 ± 0.7	*p* < 0.01 *
LLN [L]	3.09 ± 0.5	3.26 ± 0.5	*p* < 0.01 *
ULN [L]	4.59 ± 0.7	4.84 ± 0.8	*p* < 0.01 *
	**FEV1_m_** **Measured Body Height**	**FEV1_c_** **Corrected Body Height**	**FEV1_m_ vs. FEV1_c_** ***p*-Value**
Predicted value [L]	3.36 ± 0.51	3.53 ± 0.53	*p* < 0.01 *
LLN [L]	2.70 ± 0.41	2.84 ± 0.42	*p* < 0.01 *
ULN [L]	4.00 ± 0.6	4.20 ± 0.6	*p* < 0.01 *

All values are presented as a mean ± standard deviation. FVC_m_—forced vital capacity calculated for the measured body height; FVC_c_—forced vital capacity calculated for the corrected body height; FEV1_m_—forced expiratory volume in 1s calculated for the measured body height; FEV1_c_—forced expiratory volume in 1s calculated for the corrected body height; LLN—lower limit of normal; ULN—upper limit of normal. * difference statistically significant.

**Table 3 jcm-10-04877-t003:** The absolute FVC and FEV1 values, and values calculated for the measured vs. the corrected body height, *N* = 120.

Parameter	FVC 3.00 ± 0.8 L		
	FVC_m_ Measured Body Height	FVC_c_ Corrected Body Height	FVC_m_ vs. FVC_c_ *p*-Value
z-score	−1.83 ± 1.4	−2.19 ± 1.4	*p* < 0.01 *
%FVC	78.67 ± 16.7	74.62 ± 16.1	*p* < 0.01 *
	**FEV1 2.59 ± 0.7 L**		
	**FEV1_m_** **Measured Body Height**	**FEV1_c_** **Corrected Body Height**	**FEV1_m_ vs. FEV1_c_** ***p*-Value**
z-score	−1.84 ± 1.6	−2.15 ± 1.5	*p* < 0.01 *
%FEV1	77.78 ± 18.7	74.03 ± 18.2	*p* < 0.01 *

All values are presented as a mean ± standard deviation. FVCm—forced vital capacity for the measured body height; FVCc—forced vital capacity for the corrected body height; %FVC—percentage of the predicted FVC value; FEV1m—forced expiratory volume in 1s for the measured body height; FEV1c—forced expiratory volume in 1s for the corrected body height, %FEV1—percentage of the predicted FEV1 value. * difference statistically significant.

## Data Availability

The data presented in the study are available on request from the corresponding author.

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
