# Peer review of "Spirometry Examination of Adolescents with Thoracic Idiopathic Scoliosis: Is Correction for Height Loss Useful?"

_jcm, 2021, doi:10.3390/jcm10214877_

Round 1

Reviewer 1 Report

The article is very interesting and well written. Results are perfectly described in detailed tables. I only have some minor comments and suggestions:

-Line 35: ‘The aim of the study was to compare’ instead of ‘The aim of the study was to the compare’

-Material and methods section, study population: it would be nice to have a Table with the main characteristics of the patients included in the study. Some of this characteristics are mentioned in the results section and it would be nicer to have this data in the material and methods section.

-Line 140: ‘were similar and not significant’ or ‘were not significantly different’ instead of ‘were similar not significant’

Reviewer 2 Report

Abstract
The sentence "The body height loss revealed significant (p<0.05) and did impact the 12 interpretation of pulmonary parameters (p<0.01)."
does not make sense, please rewrite. The same is true for the following sentence.

Introduction
And what was the final hypothesis? You deduce to the aim of the study, but this shoud not just be descriptive. Please add a proper hypothesis 
in the end of your introduction.

Materials and Methods
The creation of Stokes' formula has been done in 2008, so what does your study has to offer in terms of novelty?
Please give a reference on why you chose the "single best" instead of the mean or average effort for the spirometry.
Kolmogorov-Smirnov was used to analyze normal distribution. So each parameter was normally distributed, since you 
after that just used the t-test?

Results
Table 1 is somehow redundant. You apply a mathematical formula to correct the body height, 
and after that it is different to the "measured body height". This does not come as a surprise.

Fig. 1: The correlation did not appear in the methods section. Why did you choose Pearson instead of Spearman?
What is the red line? If this is supposed to depict a correlation, please add the respective formula and 
a confidence interval (i.e. 5%).

"body height loss was also higher in subgroup II than in subgroup I (5.9 cm vs. 3.2
cm, p<0.01; 5.3 cm vs. 3.1 cm, p<0.01, respectively)" - in which table is this shown?

3.4:
"All the corresponding values revealed significantly different."- what was different? 
Maybe you meant "revealed significant differences"?

Please shorten the results section and focus on your main findings. Also, if reported in the text,
the differences should also appear in tables or figures.

Discussion
How big do you think a change from "mild pulmonary impairment" to "moderate impairment" would 
be for the individual patient of subgroup 2? Does this change clinical treatment?
Why "...considered in severe thoracic curvatures over 75 degress", and not any other 
curvature?

Please add limitations, like patient characteristics, just Lenke 1 or 3 idiopathic scoliosis, ...

Author Response

Dear Reviewer,

Round 2

Reviewer 2 Report

Please add the references for the "single best" value for spirometry into the manuscript, so that the reader understands this important point.

The confidence interval is still missing in Fig. 1.
